# The Frequency and Function of NKG2C^+^CD57^+^ Adaptive NK Cells in Cytomagalovirus Co-Infected People Living with HIV Decline with Duration of Antiretroviral Therapy

**DOI:** 10.3390/v15020323

**Published:** 2023-01-24

**Authors:** Khlood Alsulami, Franck P. Dupuy, Louise Gilbert, Marc Messier-Peet, Madeleine Durand, Cécile Tremblay, Jean-Pierre Routy, Julie Bruneau, Jean-Guy Baril, Benoit Trottier, Nicole F. Bernard

**Affiliations:** 1Research Institute of the McGill University Health Centre (RI-MUHC), Glen Site, Bloc E, 1001 Decarie Blvd., Room EM3.3238, Montreal, QC H4A 3J1, Canada; 2Division of Experimental Medicine, McGill University, Montreal, QC H4A 3J1, Canada; 3Infectious Diseases, Immunology and Global Health Program, Research Institute of the McGill University Health Centre, Montreal, QC H4A 3J1, Canada; 4Centre de Recherche du Centre Hospitalier de l’Université de Montréal (CRCHUM), Montreal, QC H2X 0A9, Canada; 5Department of Microbiology Infectiology and Immunology, University of Montreal, Montreal, QC H3C 3J7, Canada; 6Division of Hematology, McGill University Health Centre, Montreal, QC H4A 3J1, Canada; 7Chronic Viral Illness Service, McGill University Health Centre, Montreal, QC H4A 3J1, Canada; 8Department of Family Medicine and Emergency Medicine, Université de Montréal, Montreal, QC H3X 0A9, Canada; 9Clinique de Médecine Urbaine du Quartier Latin, Montreal, QC H2L 4E9, Canada; 10Division of Clinical Immunology, McGill University Health Centre, Montreal, QC H4A 3J1, Canada

**Keywords:** people living with HIV (PLWH), HIV, CMV, NK cells, adaptive NK cells, aging

## Abstract

Human cytomegalovirus (CMV) infection drives the expansion and differentiation of natural killer (NK) cells with adaptive-like features. We investigated whether age and time on antiretroviral therapy (ART) influenced adaptive NK cell frequency and functionality. Flow cytometry was used to evaluate the frequency of adaptive and conventional NK cells in 229 CMV^+^ individuals of whom 170 were people living with HIV (PLWH). The frequency of these NK cell populations producing CD107a, CCL4, IFN-γ or TNF-α was determined following a 6-h antibody dependent (AD) stimulation. Though ART duration and age were correlated, longer time on ART was associated with a reduced frequency of adaptive NK cells. In general, the frequency and functionality of NK cells following AD stimulation did not differ significantly between treated CMV^+^PLWH and CMV^+^HIV^-^ persons, suggesting that HIV infection, per se, did not compromise AD NK cell function. AD activation of adaptive NK cells from CMV^+^PLWH induced lower frequencies of IFN-γ or TNF-α secreting cells in older persons, when compared with younger persons.

## 1. Introduction

Human natural killer (NK) cells are involved in immune responses to viruses and tumor cells [1]. NK cells express germline encoded inhibitory and activating receptors that tune NK cell responses to target cells based on the latter’s expression of stress ligands, activating receptor ligands and loss of major histocompatibility complex (MHC) class I antigens, the ligands for inhibitory NK receptors (NKRs) [2]. NK cells, as part of the innate immune system, are primed to respond rapidly, before T and B cells can expand and differentiate into effector cells [3]. Mature NK cells comprise 10 to 15% of peripheral lymphocytes and can be divided into CD56^bright^ and CD56^dim^ subsets [4]. CD56^bright^ NK cells make up approximately 10% of circulating NK cells and are thought to be precursors of the more abundant peripheral CD56^dim^ NK cells [4,5,6,7]. As NK cells differentiate from CD56^bright^ to CD56^dim^ cells, they lose their cell surface expression of the inhibitory NKG2A NKR, sequentially acquire Killer Immunoglobulin-like Receptors (KIRs) and begin to express the maturation marker CD57 [8,9]. Most CD56^dim^ NK cells also express the activating NKR, FcγRIIIa or CD16 [1,2]. In the setting of untreated HIV infection, a poorly functional NK cell subset that is CD56^null^CD16^+^ emerges [10,11].

Human cytomegalovirus (CMV) infection is caused by a wide-spread β-herpesvirus with a prevalence of between 40 and 100% depending on age, socio-demographic factors, and geographic location [12,13]. Most HIV infected persons are also CMV co-infected [14]. Prior to the availability of antiretroviral treatment (ART), CMV co-infection of People Living with HIV (PLWH) was considered to be an opportunistic infection associated with important morbidity and mortality [15,16]. In the era of ART availability, CMV and HIV co-infections worsen each other’s disease course by contributing to systemic inflammation, cardiovascular disease and immune senescence [17,18,19,20,21,22,23]. CMV infection reshapes the NK cell repertoire by driving the expansion of a subset of NK cells expressing NKG2C and CD57 [22,23,24,25,26,27]. NKG2C is an activating receptor belonging to the C-type lectin receptor family, which is expressed as a heterodimer with CD94 [28]. The ligand for NKG2C/CD94 is HLA-E, whose cell surface expression is stabilized by epitopes derived from the leader sequence of HLA-A, -B, -C and -G antigens or the UL40 CMV gene product [23,24,25,28,29,30,31]. These NKG2C^+^CD57^+^ NK cells have memory like features such as antigen-specific clonal expansion following CMV infection and form long-lived memory cells, which are features of adaptive lymphocytes. These cells also exhibit epigenetic changes similar to those observed in cytotoxic CD8 T cells, including DNA methylation-dependent silencing of the promyelocytic leukemia zinc transcription factor (PLZF) and stochastic loss of expression of signaling molecules such as FcεRIγ, spleen tyrosine kinase and EWS/FLI1-activated transcript 2 [32,33,34,35]. These NKG2C^+^CD57^+^ NK cells are called adaptive NK cells. Adaptive NK cells differ from conventional NK cells by expressing lower frequencies of the Natural Cytotoxicity Receptors (NCRs) NKp30 and NKp46, CD161, higher frequencies of inhibitory KIRs, particularly those using HLA-C as ligands and the leukocyte immunoglobulin-like receptor family, member 1 (LILRB1), and similar levels of CD57, NKG2D and CD16 [24,36,37].

Several studies have compared the frequency of NKG2C^+^CD57^+^ adaptive NK cells in CMV^+^PLWH and in CMV mono-infected (CMV^+^HIV^−^) individuals [25,38,39,40]. While three of these studies found that HIV infection further accentuated adaptive NK cell expansion observed in CMV mono-infected persons, Guma et al. found no between-group differences in the frequency of adaptive NK cells. The possibility that the age of the study subjects or how long they had been on ART may account for the presence or absence of differences in adaptive NK cell frequencies between groups has not been addressed and could not be assessed as precise information on the mean/median age and age range of the study populations analyzed was only provided in two of these studies [25,38]. Information on how long they had been on ART was often not specified.

Herein, we screened participants of the Canadian HIV and Aging Cohort Study (CHACS), all aged > 40 yrs and on ART for a median of 16 yrs, who were CMV^+^PLWH and CMV-mono-infected individuals for the frequency of their adaptive NK cells. We then compared the frequency of these cells in CHACS participants with those in CMV^+^PLWH and CMV-mono-infected individuals who were younger than 40 yrs of age. We also investigated the capacity of NKG2C^+^CD57^+^ adaptive NK cells from CMV^+^PLWH and CMV mono-infected subjects who were below versus above 40 yrs of age to degranulate and produce cytokines/chemokines following antibody dependent (AD) stimulation. The frequency of adaptive NK cells did not differ significantly between these CMV^+^PLWH and CMV mono-infected populations in those who were ≥40 yrs of age and treated for a median of 16 yrs but was higher in CMV^+^PLWH than in CMV^+^HIV^-^ individuals who were <40 yrs of age and treated for a median of 1.4 yrs, consistent with an inverse correlation between the frequency of adaptive NK cells and either age and time on ART in CMV^+^PLWH. When the frequency of adaptive NK cells was compared in CMV^+^PLWH who were <40 versus >40 yrs of age on ART for a similar length of time, no significant differences were observed, suggesting that time on ART was more important than age as a determinant of NK cell frequency. Older age, which correlated with length of time on ART, was associated with a lower frequency of adaptive NK cells secreting IFN-γ and TNF-α following AD stimulation. 

## 2. Materials and Methods

### 2.1. Ethics Statement

This research study was approved by the Research Ethics Boards of the Centre Hospitalier de l’Université de Montréal and the McGill University Health Centre (Project Identification Code 2019-4605). It was conducted according to the principles expressed in the Declaration of Helsinki. Written informed consent was obtained from each study subject for the collection of specimens and subsequent analyses.

### 2.2. Study Population

The study population included 229 CMV seropositive individuals. Study population characteristics are shown in Table 1. Of the 50 individuals who were below 40 yrs of age, 28 were CMV^+^PLWH enrolled in the Montreal Primary Infection (PI) cohort and 22 were HIV uninfected CMV mono-infected individuals. All of the 164 individuals enrolled in the CHACS were ≥40 yrs of age, which was one of the inclusion criteria for this cohort. Of these, 127 were CMV^+^PLWH, and 37 were CMV mono-infected individuals. An additional 15 individuals, enrolled in the Montreal PI cohort were >40 yrs of age. The study design and protocol of the Montreal PI Cohort and the CHACS have been previously reported [41,42]. Of the subjects enrolled in the CHACS, 84% were male and 85% were Caucasian. In the Montreal PI cohort 96% were male and 89% were Caucasian. Subjects enrolled in the Montreal PI cohort included individuals recruited within the first 6 months of HIV infection, who were then followed an average of every 3 months for up to 4 years. At each clinic visit, blood was drawn for isolation of plasma and peripheral blood mononuclear cells (PBMC), which were stored frozen until use [43]. The PI cohort samples used in this study were from time points collected during the chronic phase of infection, a median (interquartile range [IQR]) of 2.2 (1.5, 2.9) yrs after their presumed date of infection at which time participants had been on ART for a median of 1.7 (1.2, 2.05) yrs with viral loads <50 copies/mL of plasma. The younger group of CMV mono-infected individuals were 33.2 (30.8, 38.2) yrs of age. CHACS participants were recruited from HIV and sexually transmitted disease clinics in Montreal, Quebec, Canada. Most were men who have sex with men. All CMV^+^PLWH were on ART for a median of 16 (8.6, 19.1) yrs and had viral loads of <50 copies/mL of plasma. 

### 2.3. CMV Testing

Plasma samples collected either at the same time point or at a time before the time point used for the phenotypic and functional assessments described in this report were collected from each participant to assess their CMV serological status. This was done by using CMV IgG enzyme immunoassays (EIA) from either CD Creative Diagnostics (Shirley, NY, USA), Abcam (Cambridge, MA, USA), GenWay, Biotech LLC (San Diego, CA, USA) or Abbott Diagnostics (AxSym CMV or Architect CMV IgG, Abbott Park, IL, USA). Testing for CMV antibodies was done according to the manufacturers’ directions.

### 2.4. Staining PBMC for Adaptive NK Cells

Frozen PBMCs were thawed and resuspended in RPMI 1640; 5% fetal bovine serum (FBS); 2 mM L-glutamine; 50 international units/mL penicillin; 50 µg/mL streptomycin (R5) (all from Wisent, Inc., Saint Jean-Baptiste, QC, Canada). LIVE/DEAD fixable dead cell stain (Invitrogen, Saint Laurent, QC, Canada) was added to the PBMCs as per the manufacturer’s directions before the cells were surface stained using a panel that included the following fluorochrome conjugated antibodies to CD3-BV785 (clone OKT3), CD14-BV785 (M5E2), CD19-BV785 (HIB19), CD56-BV605 (HCD56) (all from BioLegend, San Diego, CA, USA), CD57-PE (TB01) (Life Technologies, Burlington, ON, Canada), CD16-APC-Cy7 (3G8) (BD Bioscience, Baltimore, MD, USA), NKG2C-PE-Vio770 (REA205) and NKG2A-APC (REA110) (Miltenyi Biotec, Auburn, CA, USA) for 20 min at 4 °C.

### 2.5. AD NK Cell Activation (ADNKA) Assay

The ADNKA assay was used to assess the frequency of NK cells producing the degranulation marker CD107a, the chemokine CCL4 and the cytokines IFN-γ and TNF-α by adaptive NK cells and conventional NK cells from the study subjects following stimulation of PBMC with HIVIG opsonized, sorted, infected CEM.NKR.CCR5 (siCEM) cells. HIVIG is a pool of purified IgG from asymptomatic PLWH with CD4^+^ counts above 400 cells/μL, (HIVIG was obtained from the National Agri-Food Biotechnology Institute (NABI) and the National Heart, Lung, and Blood Institute (NHLBI) through the NIH AIDS Reagent Program, Division of AIDS (DAIDS), National Institutes of Allergy and Infectious Diseases (NIAID), National Institutes of Health (NIH) [44]. HIVIG contains antibodies recognizing the HIV Envelope expressed on HIV infected cells [45,46]. CEM cells were obtained from the NIH AIDS Reagent Program, DAIDS, NIAID, NIH as CEM.NKR.CCR5 cells from Dr. Alexandra Trkola [47,48]. 

SiCEM cells are HIV infected CEM cells. The virus used to infect CEM cells is an NL4.3 based HIV virus pseudotyped with Bal-Envelope and engineered to express the murine Heat Stable Antigen (HSA or mCD24) [49]. After infection, these cells were sorted for HSA expression and expanded in culture. SiCEM cells were virtually 100% HIV infected as determined by expression of HSA and HIV p24 and expressed no cell surface CD4, as it was downmodulated on infected cells by wild type Nef and Vpu [50,51]. The absence of cell surface CD4 precludes the cell surface HIV Envelope from adopting its open conformation [45]. Thus, siCEM cells expose the HIV Env in a closed conformation, analogous to what would be present on genuinely HIV infected cells. 

Study subject PBMCs were used as responder (R) cells in the ADNKA assay. Frozen PBMCs were thawed and rested for 2 h in R5 media. One million stimulatory (S) siCEM cells were opsonized with 50 µg/mL of HIVIG in a volume of 100 μL of RPMI 1640; 10% FBS; 2 mM L-glutamine; 50 international units/mL penicillin; 50 µg/mL streptomycin (R10) for 20 min at a room temperature (RT) in V-bottomed 96-well tissue culture plates. Control siCEM S cells remained un-opsonized. One hundred μL of PBMCs (R) at a concentration of 10^7^ cells/mL of R10 were added to the wells containing HIVIG or un-opsonized siCEM cells and co-cultured at 37 °C, in a humidified, 5% CO_2_ incubator for 6 h with BV711 conjugated anti-CD107a (clone M4A3: BioLegend) Ab. After 1 h, Golgi stop (monensin) and Golgi plug (brefeldin A) (both from BD Bioscience) were added according to the manufacturers’ instructions for the remaining 5 h of culture. Cells were washed in RPMI 1640; 2% FBS before staining for the same cell surface markers as used for quantifying the frequency of adaptive NK and conventional NK cells. After cell surface staining, cells were permeabilized (Fixation/Permeabilization Kit, BD Bioscience) as per manufacturer directions and stained intra-cellularly with Abs to CCL4-AF700 (clone D21-1351), IFN-γ-BV510 (clone B27) and TNF-α-BV650 (clone Mab11, all from BD Bioscience) for 30 min at room temperature in the dark. Cells were then washed and resuspended in 2% paraformaldehyde. 

### 2.6. Flow Cytometry

A total of 1.5 × 10^6^ to 1.8 × 10^6^ events were acquired for each sample using an LSR Fortessa instrument (BD Bioscience, San Jose, CA, USA). Results were analyzed using FlowJo^TM^ software v10.3 (BD, Ashland, OR, USA). NK cell subsets were identified as live, singlet, CD3^-^CD14^-^CD19^-^CD56^dim^ cells. Adaptive NK cells were defined as NKG2C^+^CD57^+^CD56^dim^ NK cells; conventional NK cells were defined as NKG2C^-^CD56^dim^ NK cells. Florescence minus one staining was used to set gates for each experiment. Single-stained beads (BD^TM^ CompBead, BD Bioscience) were used to set compensation. Functional markers were gated on within the adaptive NK or conventional NK cell gates.

### 2.7. Statistical Analysis

Statistical analysis and graphical presentation of results were performed using GraphPad Prism 9.3.1 (GraphPad Software Inc., La Jolla, CA, USA). The statistical significance of differences between two unmatched and two matched groups was assessed using two-tailed Mann—Whitney tests and Wilcoxon tests, respectively. The significance of correlations between the frequency of NKG2C^+^CD57^+^ adaptive NK cells and age or ART duration was assessed using Spearman’s correlation tests. *p*-values of <0.05 were considered significant. In cases where multiple comparisons were made, Bonferroni corrections were applied. 

## 3. Results

### 3.1. Study Population Characteristics 

Of the 229 individuals screened for the frequency of their adaptive NK cells, 170 (74.2%) were PLWH and 59 (25.8%) were CMV mono-infected. All PLWH were receiving ART. Two hundred and two (88.2%) of the study participants were male. Table 1 provides information of the age, sex distribution, duration of infection for the 2 groups of subjects enrolled in the PI cohort, VL, CD4 count CD4:CD8 and years on ART for the groups who were PLWH. The significance of between-group differences in these parameters is provided in the table.

### 3.2. The Frequency of NKG2C^+^CD57^+^CD56^dim^ Adaptive NK Cells Is Dependent on the Duration of ART 

We first evaluated the frequency of NKG2C^+^CD57^+^CD56^dim^ adaptive NK cells in CMV^+^PLWH and CMV mono-infected individuals enrolled in the CHACS. Figure 1A shows the strategy used to gate on adaptive NK and conventional NK cells, which were identified as live, singlet, lymphocytes, which were CD3^-^CD14^-^CD19^-^CD56^dim^. There was no significant between-group difference in the frequency of NKG2C^+^CD57^+^CD56^dim^ adaptive NK cells in CMV^+^PLWH and CMV mono-infected CHACS subjects aged >40 yrs (16.1% (6.6, 39.0] vs. 13.7% (4.3, 33.4], respectively, *p* = 0.74, Mann—Whitney test) (Figure 1B). Some previous studies had reported that CMV^+^PLWH had higher frequencies of adaptive NK cells than CMV mono-infected individuals [38,39,40]. In two of these reports, the age range of the populations studied was not reported [39,40] while in Heath et al., the median age of the CMV^+^PLWH and CMV mono-infected populations overlapped that of the participants in the CHACS [38]. We questioned whether age may affect the frequency of adaptive NK cells and whether the differences between our results and those generated by others with respect to between-group differences in adaptive NK cell frequencies was due to differences in age. When we screened CMV^+^PLWH and CMV mono-infected individuals aged <40 yrs for the frequency of their adaptive NK cells, we found that their frequency was significantly higher in CMV^+^PLWH than in CMV mono-infected subjects (39.8% (18.8, 58.2) vs. 15.6% (2.5, 33.3), *p* = 0.003, Mann—Whitney test) (Figure 1C). Furthermore, the frequency of adaptive NK cells was inversely correlated with age for all CMV^+^PLWH (Groups 1 and 3) (r = −0.18, *p* = 0.02, Spearman correlation test) (Figure 1D). While a similar inverse correlation between the frequency of adaptive NK cells and age was noted for CMV mono-infected individuals (Groups 2 and 5), this observation did not achieve statistical significance (Figure 1E).

Figure 2 shows that the frequency of adaptive NK cells was significantly higher in CMV^+^PLWH who were <40 than compared to ≥40 yrs of age (39.8% [21.5, 54.9] versus 16.1% [6.64, 39.0], respectively, *p* = 0.002, Mann—Whitney test). In CMV mono-infected participants, the frequency of adaptive NK cells did not differ in those <40 versus >40 yrs of age (15.6% [3.2, 31.3] versus 12.3% [5.1, 31.5], respectively, *p* = 0.88, Mann—Whitney test). These results show that the absence of a significant differences in the frequency of adaptive NK cells in CMV^+^PLWH compared to CMV mono-infected persons aged ≥40 yrs was due to a decline in the frequency of these cells with age in CMV^+^PLWH. It is notable that length of time on ART is likely to increase with duration of infection. The presumed date of infection was not available for CHACS participants, precluding the possibility of estimating the duration of infection. However, duration of time on ART was available for CMV^+^PLWH who were participants in the CHACS and PI cohort participants. There was a significant negative correlation between the frequency of adaptive NK cells and time on ART (r = −2, *p* = 0.009, Spearman correlation) (Figure 1F). 

To address whether age or time on ART was a more important determinant of changes in the frequency of adaptive NK cells, we examined the frequency of these cells in 15 PI cohort participants who were >40 yrs of age but on ART for 1.94 (1.8, 2.0) yrs. In terms of age, these older PI subjects did not differ from CHACS participants but were older than the PI cohort individuals who were <40 yrs old (Figure 3A). In terms of time on ART, the two PI groups did not differ from each other and both were younger than the CHACS participants (Figure 3B). The frequency of adaptive NK cells did not differ in the PI cohort subjects who were less than versus greater than 40 yrs of age. The percent of these cells was higher in younger PI subjects than in CHACS participants and trended toward being higher in older PI subjects than in CHACS participants (Figure 3C). Together, these results support the interpretation that increasing length of time on ART, rather than increasing age is associated with declining frequencies of adaptive NK cells. Furthermore, the absence of an effect of age on the frequency of adaptive NK cells in CMV mono-infected individuals (Figure 1E and Figure 2B) would support this conclusion as they are ART naïve.

### 3.3. Adaptive NK and Conventional NK Cell Function in CMV^+^PLWH and CMV Mono-Infected Subjects

We next compared the ability of adaptive and conventional NK cells from CMV^+^PLWH and CMV mono-infected subjects to respond to an AD stimulus. Figure 1A shows the gating strategy used to determine the frequency of adaptive NK (right-hand panel, upper right-hand quadrant) and conventional NK (right-hand panel, combined lower left- and right-hand quadrants) cells that degranulated or secreted CCL4, IFN-γ or TNF-α following stimulation with HIVIG opsonized siCEM cells. 

The frequency of adaptive NK (Figure 4B–E) and conventional NK (Figure 3F–I) cells producing CD107a, IFN-γ and TNF-α did not differ significantly when from CMV^+^PLWH or CMV mono-infected subjects (Mann—Whitney). Although the frequency of adaptive NK, but not conventional NK cells producing CCL4 was higher or showed a trend towards being higher in CMV-mono-infected persons compared to CMV^+^PLWH, the difference fell below the level of significance when a Bonferroni correction was applied. The significance of this observation was that it shows that the frequency of functional adaptive NK cells persists in the setting of treated HIV infection. 

A comparison of within-individual frequencies of functional adaptive and conventional NK cells responding to stimulation with HIVIG opsonized siCEM cells showed that a higher frequency of adaptive NK than conventional NK cells from CMV^+^PLWH expressed CD107a, secreted IFN-γ or TNF-α (*p* < 0.0001 for all, Wilcoxon matched-pairs test) (Appendix A). Additionally, a higher frequency of adaptive than cNK cells from CMV mono-infected individuals secreted CCL4, IFN-γ and TNF-α, though the statistical significance of between-group differences in the frequency of CCL4 secreting cells fell below the level of significance when a Bonferroni correction was applied (Appendix A). 

### 3.4. Effect of Age and Time on ART on NKG2C^+^CD57^+^CD56^dim^ Adaptive NK Cell Functions 

Figure 5 compares the frequency of adaptive NK cells expressing CD107a and secreting CCL4, IFN-γ and TNF-α in response to stimulation with HIVIG opsonized siCEM cells in participants aged <40 versus ≥40 yrs of age. There were no significant differences in the frequency of antibody opsonized target cell stimulation of adaptive NK cells expressing CD107a or secreting CCL4 from either CMV^+^PLWH or CMV mono-infected subjects who were <40 versus ≥40 yrs of age (Figure 5A,B,E,F). A significantly higher frequency of adaptive NK cells from younger than older CMV^+^PLWH secreted IFN-γ and TNF-α (*p* ≤ 0.005 for both, Mann—Whitney tests, Figure 5C,D). These differences remained statistically significant following the application of Bonferroni corrections. On the other hand, the frequency of adaptive NK cell producing each of the functions tested did not differ significantly in CMV mono-infected subjects (Figure 5G,H). 

Appendix A shows how the frequency of functional adaptive NK cells correlates with age (Appendix A) and duration of ART (Appendix A) in CMV^+^PLWH and with age in CMV^+^HIV^-^ individuals (Appendix A). The frequency of adaptive NK cells secreting IFN-γ and TNF-α are inversely correlated with age (Appendix A) and with duration of time on ART (Appendix A) in CMV^+^PLWH. There is a nonsignificant trend towards the frequency of CCL4 secreting cells correlating with time on ART in CMV^+^PLWH that disappears upon the application of a Bonferroni correction. No correlations between the frequency of AD stimulated functional adaptive NK cells and age achieve statistical significance in CMV mono-infected persons.

## 4. Discussion

We showed here that the frequency of adaptive NK cells in individuals younger than 40 yrs of age was significantly higher in CMV^+^PLWH than in CMV mono-infected individuals, while in participants who were 40 yrs of age or older there was no significant between-group difference in the frequency of adaptive NK cells. We also observed a negative correlation between the frequency of adaptive NK cells with increasing age in CMV^+^PLWH. As age and time on ART are positively correlated with each other we investigated whether age or time on ART could account for changes in the frequency of adaptive NK cells. In two groups of study subjects on ART for a similar duration we found no differences in the frequency of adaptive NK cells despite significant differences in age. This finding suggested that increasing time on ART, rather than increasing age, was associated with declining frequencies of adaptive NK cells. We observed no significant differences in the frequency of adaptive and conventional NK cells from CMV^+^PLWH and CMV mono-infected individuals responding to AD stimulation by externalizing CD107a, or secreting CCL4, IFN-γ or TNF-α. Thus, HIV infection was not associated with a reduced frequency of functional adaptive, compared to conventional, NK cells in CMV seropositive persons responding to AD stimulation. Indeed, a within-individual comparison of the AD functionality of adaptive versus conventional NK from CMV^+^PLWH showed higher frequencies of functional adaptive than conventional NK cells for all functions tested, except CCL4 secretion. This was also the case for adaptive NK cells from CMV mono-infected individuals for secretion of IFN-γ and TNF-α. The frequency of adaptive NK cells from CMV^+^PLWH responding to AD stimulation by secreting IFN-γ or TNF-α was higher in younger than older individuals who also differ in length of time on ART. On the other hand, the frequency of functional adaptive NK cells from CMV mono-infected persons responding to AD stimulation did not differ significantly in participants from these two age groups. This finding is consistent with time on ART influencing adaptive NK cell frequency and functionality since CMV mono-infected individuals are HIV uninfected and thus not on ART. 

We initially studied CHACS participants who were all ≥40 yrs of age. Comparing the frequency of adaptive NK cells in CMV^+^PLWH and CMV mono-infected persons revealed no significant between-population differences. The frequency of adaptive NK cells in CMV^+^PLWH versus CMV mono-infected persons had been examined by others and found to be higher in some (35–37), but not in all (23) reports. This prompted us to recruit CMV^+^PLWH and CMV mono-infected persons who were <40 yrs of age so that we could compare results with those who were ≥40 yrs of age. In work reported by others, either the median (IQR or range) or mean ± standard deviation of the ages of the PLWH and HIV uninfected study populations examined was often not provided. In only one instance was the age distribution reported for the subset of study participants who were CMV^+^PLWH [38]. Heath et al. showed that the frequency of adaptive NK cells was higher in CMV^+^PLWH than in CMV mono-infected individuals who had a median (IQR) age of 49 (45, 55) for CMV^+^PLWH and 48 (39, 61) for CMV mono-infected persons [38]. This observation differs from the results reported here where CHACS participants who were also ≥40 yrs of age had adaptive NK cell frequencies that did not differ significantly between CMV^+^PLWH and CMV mono-infected participants. The reasons underlying this discrepancy may be due to differences in time on ART rather than age being a more important determinant of the frequency of adaptive NK cells. In Heath et al., most PLWH were reported to be on ART, though neither the duration of HIV infection nor the time on ART was specified. In the PLWH included in this study those aged ≥40 yrs had been on ART for a median (IQR) of 16 (8.6, 19.1) yrs while those who were <40 yrs of age were more recently infected and therefore on ART for a shorter duration of 1.4 (1.0–2.2) yrs. By including individuals of younger age and length of time on ART in this study, we built upon previously reported results by showing that there was an inverse correlation between the frequency of adaptive NK cells and both age and time on ART in CMV^+^PLWH. The correlation between adaptive NK cell frequency and age for CMV mono-infected persons was more modest and did not achieve statistical significance. While this could be due to the smaller samples size of the CMV mono-infected population compared to the CMV^+^PLWH group, if ART duration is important in influencing adaptive NK cell frequency, the absence of a correlation between this variable and age in CMV mono-infected persons would be consistent with their HIV seronegative status. A limitation of this analysis is its cross-sectional nature. Work done by others showed that in CMV^+^PLWH starting ART in the chronic phase of infection, the frequency of adaptive NK cells, as defined by being positive for cell surface NKG2C or negative for intracellular FcεRIγ, remained stable for up to 24 months on ART [31,39]. The frequency of NKG2C^+^ adaptive NK cells in a small sample of five individuals examined longitudinally in untreated early infection, established viremic infection and after 1 yr on ART was stable over these three time points [34]. In a fourth study, ART treatment of CMV^+^PLWH for 48 weeks was accompanied by an increase in the frequency of NKG2C^+^CD57^+^ adaptive NK cells from pre-ART initiation to 48 weeks on ART time points [52]. While these longitudinal studies provide information on the within-individual changes with time in the frequency of adaptive NK cells over time intervals of up to 2 yrs on ART, they do not address changes that may occur over longer periods of time such as after a decade or more on ART and are limited by the small number of subjects studied. To our knowledge, this is the first study to examine frequencies of adaptive NK cells in persons on ART for various length of time and in some cases beyond 20 years on ART. 

CMV infection is accompanied by several epigenetic modifications [32,33]. For example, adaptive NK cells lack the intracellular signaling adaptor molecule FcεRIγ, due to hypermethylation of the *FCER1G* promoter, which silences this gene encoding FcεRIγ [32,33,35,53,54,55]. FcεRIγ is a molecule that contains two immunoreceptor tyrosine-based activation motifs (ITAMs) while the intracellular signaling adaptor molecule, CD3ζ has three ITAMs. Both molecules are present in conventional NK cells, where they usually form heterodimers or FcεRIγ homodimers, which participate in the transmission of signals from the activating NKR CD16 [56,57,58]. When FcεRIγ is absent, as occurs in many adaptive NK cells, the CD16 signaling cascade uses CD3ζ homodimers whose six ITAMs support more robust CD16 Fc receptor mediated effector responses, favoring adaptive NK cells achieving broader and more potent antigen specificity through AD functions compared to conventional NK cells [56,59]. The adaptive NK cell subsets characterized as NKG2C^+^ and FcεRIγ^—^are largely overlapping, though not identical [60]. It should be noted that NKG2C expression appears to not be essential for conferring adaptive NK cells with superior AD responsiveness compared to conventional NK cells [56]. In CMV infected individuals homozygous for a deletion mutant of *NKG2C* who fail to express this receptor, there exist NK cells having the same features as adaptive NK cells including exhibiting a terminal differentiation phenotype, functional reprogramming and epigenetic modifications similar to those seen in NKG2C^+^ or FcεRIγ^—^adaptive NK cells [56]. The NKR CD2 can synergize with NKG2C in typical adaptive NK cells and plays an important role in the AD responses of adaptive NK cells even when NKG2C is absent on these cells [56]. 

To investigate the effect of HIV infection on the AD function of adaptive NK cells, we enumerated the frequency of adaptive NK and conventional NK cells externalizing CD107a and secreting CCL4, IFN-γ and TNF-α following AD stimulation. While the frequency of functional adaptive and conventional NK cells tended to be modestly lower in CMV^+^PLWH than in CMV mono-infected individuals, between-population differences did not achieve statistical significance. We also performed a sub-analysis after separating study subjects into groups who were <40 and ≥40 yrs of age. In both cases, between-population differences in the frequency of adaptive NK and conventional NK cells producing the four functions tested did not differ significantly (not shown). This suggests that treated HIV infection has a minimal effect on the responsiveness of adaptive compared to conventional NK cells to AD stimulation. 

This contrasts with a study conducted in rhesus macaques (RM). Both adaptive NK cells, which were characterized as FcεRIγ^—^, and conventional NK cells from rhesus CMV^+^ (rhCMV^+^) RMs responded robustly and similarly to AD stimulation by externalizing CD107a and secreting CCL4, IFN-γ or TNF-α. The adaptive NK cells were shown to preferentially use the CD3ζ/ZAP70 CD16 signaling pathway rather than the FcεRIγ/Syk signaling pathway for adaptive NK cell activation. However, adaptive and conventional NK cells in rhCMV^+^ RMs co-infected with SIV had compromised AD functionality [59]. When the CD16 signaling pathway of conventional NK and adaptive NK cells from rhCMV^+^ and rhCMV^+^SIV^+^ NK cells was compared, the CD3ζ/ZAP70 CD16 signaling pathway in rhCMV^+^SIV^+^ animals was compromised compared to that of rhCMV^+^ RMs [59]. While it is possible that host species and pathogen differences could account for compromised adaptive NK cell function in RMs that was not observed in adaptive NK cells from CMV^+^PLWH, it is also possible that in the human participants studied here, ART, which suppressed viral load, preserved adaptive and conventional NK cell functionality in the setting of HIV infection. 

A higher frequency of within-individual NKG2C^+^CD57^+^ adaptive compared to conventional NK cells from CMV^+^PLWH and CMV-mono-infected individuals responded to stimulation through CD16 by secreting IFN-γ and TNF-α. This was also the case for the frequency of adaptive NK cells externalizing CD107a in CMV^+^PLWH. The superior ability of adaptive, compared to conventional, NK cells to respond to AD stimulation by secreting IFN-γ and TNF-α has been reported by others [56,61,62]. The mechanism underlying this phenomenon is likely due to the promoter regions of *IFNG* and *TNF*, the genes encoding IFN-γ and TNF-α, respectively, being hypomethylated in adaptive NK cells allowing for greater cytokine production upon AD stimulation through ITAM-coupled receptors such as CD16 expressed on adaptive NK cells [32,61,62]. Taken together, epigenetic modification is responsible for several specific phenotypes of adaptive NK cells. This is likely to be the mechanism underlying the elevated within-individual capacity of adaptive, compared to conventional, NK cells to release these cytokines following activation via CD16. These results provide further support for the interpretation that HIV infection had little impact on the AD function of adaptive NK cells over that seen for conventional NK cells.

This study has some limitations. The population size, particularly for CMV mono-infected subjects, was small, which may have precluded achieving significance for some of the analyses performed in which only CMV-mono-infected subjects were included. The results presented have distinguished data generated by cells originating from males and females by color. The majority of study subjects were male and Caucasian, as would be expected given that study subjects recruited to both the CHACS and the PI cohorts were male and Caucasian. This precluded performing analyses on women only or on races/ethnicities other than Caucasian due to the small size of these groups. Overall, it appears that the data attributed to women appear similarly distributed to that of men. 

Adaptive NK cells likely play a role in CMV control through interactions between NKG2C and HLA-E CMV encoded UL40 peptide complexes on CMV infected cells that activate these cells. Adaptive NK cells may also exert control over HIV and other infections that induce pathogen specific antibodies through their ability to become activated by ADNKA. Our observation that time on ART decreases the frequency of adaptive NK cells and their functionality may have implications in terms of reduced control of CMV, HIV and other pathogens. The results presented in this manuscript would be relevant for HIV infected individuals who are treated with ART long term though the effect may not be limited to HIV. Further investigations are warranted to explore how the reduced frequency and function of adaptive NK cells with time on ART impacts on human health and pathogen control. 

## 5. Conclusions

These results provide evidence supporting time on ART-associated changes in the frequency and function of NKG2C^+^CD57^+^ adaptive NK cells in participants who were CMV^+^PLWH. These ART-related changes were evident in CMV^+^PLWH. Our results showed, for the first time, a reduction in both NKG2C^+^CD57^+^ adaptive NK cell frequency and function accompanied with time on ART in CMV^+^PLWH. Adaptive and conventional NK cell function was not impaired in the presence of HIV infection. Furthermore, a decreased frequency of functional adaptive NK cells was associated with increasing age and time on ART, and an augmented population of NKG2C^+^CD57^+^ adaptive NK producing IFN-γ and TNF-α was found in younger CMV^+^PLWH.

## Figures and Tables

**Figure 1 viruses-15-00323-f001:**
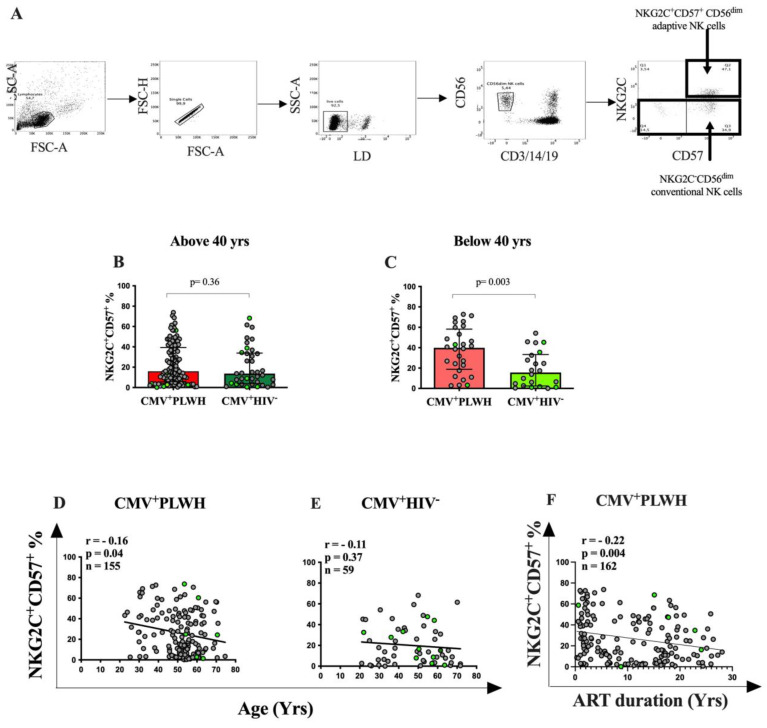
The frequency of adaptive NK cells in CMV^+^PLWH and CMV mono-infected persons declines with age and time on antiretroviral therapy (ART). (**A**) Gating strategy to detect adaptive and conventional NK cell frequencies. Live, singlet lymphocytes were gated on from peripheral blood mononuclear cells. From these, NK cells were identified as CD3^-^CD14^-^CD19^-^CD56^dim^ cells. The frequency of adaptive and conventional NK cells was determined as the proportion of NKG2C^+^CD57^+^CD56^dim^ NK cells (upper right-hand quadrant) and NKG2C^-^CD56^dim^ NK cells (combined lower left- and right-hand quadrants) in the CD56^dim^ NK cell gate, respectively. (**B**,**C**) The y-axes show the frequency of NKG2C^+^CD57^+^ adaptive NK cells in CMV^+^PLWH and CMV mono-infected persons aged >40 yrs (**B**) and <40 yrs (**C**). Each data point represents results for a single individual. Bar heights and error bars show the median and interquartile ranges (IQR) of the frequency of adaptive NK cells for each group. Mann—Whitney tests were used to assess the statistical significance of between-group differences for the frequency of adaptive NK cells. *p*-values indicating the significance of between group differences are shown over the lines linking the groups being compared. Bars for CMV+PLWH individuals are colored in red whereas bars for CMV+HIV- individuals are colored in green. Dark red and dark green represent subjects above 40 yrs, while light red and light green represent subjects below 40 yrs. (**D**–**F**) Correlations between the frequency of adaptive NK cells and study subject age (**D**,**E**) or duration of time on ART (**F**). The y-axes show the frequency of NKG2C^+^CD57^+^ adaptive NK cells while the x-axes show the age in years of CMV^+^PLWH (**D**), CMV mono-infected persons (**E**) or time on ART in years for CMV^+^PLWH. Spearman correlation tests were used to assess the statistical significance of these correlations. The correlation coefficients (r), *p*-values, and the number of subjects tested in each panel are shown in the top left corner of each graph. Data points corresponding to results attributed to females are distinguished from those attributed to males by being shown in light green.

**Figure 2 viruses-15-00323-f002:**
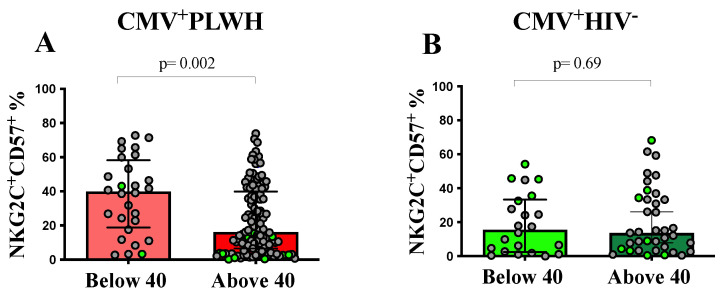
CMV^+^PLWH who are older than 40 yrs of age have lower frequencies of adaptive NK cells than those who are younger than 40 yrs of age. The y-axes show the frequency of NKG2C^+^CD57^+^CD56^dim^ NK cells in (**A**) CMV^+^PLWH and (**B**) CMV mono-infected (CMV^+^HIV^−^) subjects who are <40 and >40 yrs of age. Each data point represents a single individual. Bar graph heights and error bars show the median and IQR for the frequency of adaptive NK cells for each group. Bars colors identify the same groups as in Figure 1 panel B and C. Mann—Whitney tests were used to assess the statistical significance of between-group differences for the frequency of adaptive NK cells. *p*-values indicating the significance of between group differences are shown over the lines linking the groups being compared. Data points corresponding to results attributed to females are distinguished from those attributed to males by being shown in light green.

**Figure 3 viruses-15-00323-f003:**
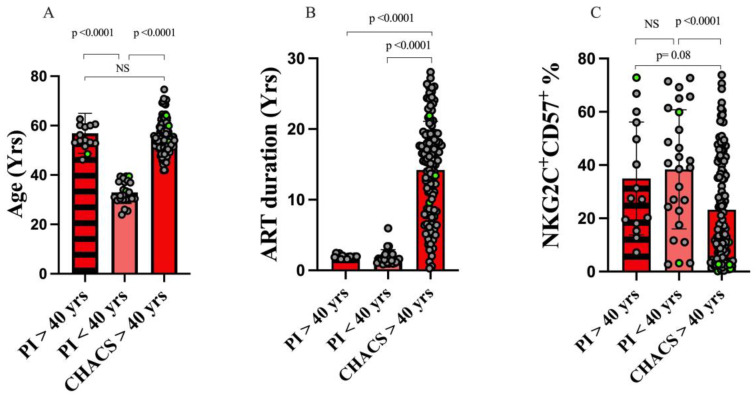
Time on ART rather than age is associated with lower frequencies of adaptive NK cells. (**A**–**C**) The y-axes show the age (**A**), time on ART (**B**) and the frequency of adaptive NK cells (**C**) for 15 CMV^+^PLWH from the HIV Primary Infection (PI) cohort who were >40 yrs of age and on ART for a median of 1.9 yrs, 26 CMV^+^PLWH from the HIV PI cohort who were <40 yrs of age and on ART for a median of 1.4 yrs and 126 participants of the Canadian HIV and Aging Cohort Study (CHACS) who were >40 yrs of age and on ART for a median of 16 yrs. Each data point represents results for a single individual. Bar heights and error bars show the median and interquartile ranges (IQR) for each data set. Kruskal—Wallis tests with Dunn’s post tests were used to assess the significance of between-group differences. *p*-values indicating the significance of between group differences are shown over the lines linking the groups being compared. Data points corresponding to results attributed to females are distinguished from those attributed to males by being shown in light green. PI = primary infection; CHACS = Canadian HIV and Aging Cohort Study; yrs = years; NS = not significant.

**Figure 4 viruses-15-00323-f004:**
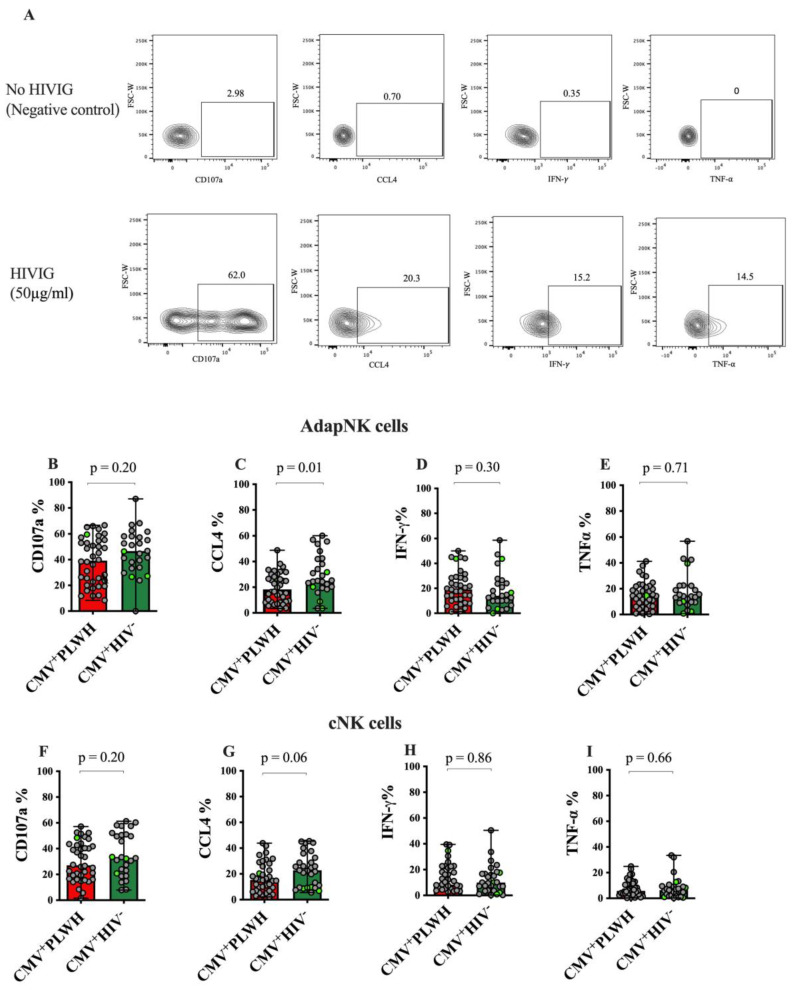
HIV infection does not compromise the functionality of adaptive and conventional NK cell responses to antibody opsonized HIV infected cells. (**A**) Gating strategy to detect the frequency of functional adaptive and conventional NK cells. After stimulating peripheral blood mononuclear cells for 6 hrs with anti-HIV Envelope-specific antibody opsonized or un-opsonized HIV infected cells (siCEM) and anti-CD107a specific antibody, cells were surface stained and stained intracellularly with antibodies to CCL4, IFN-γ and TNF-⍺. Adaptive and conventional NK cells were gated on as shown in Figure 1A. The frequency of adaptive NK cells (**B**–**E**) and conventional NK cells (**F**–**I**) producing CD107a (**B**,**F**), CCL4 (**C**,**G**), IFN-γ (**D**,**H**) and TNF-⍺ (**E**,**I**) was assessed. The condition in which PBMC were stimulated with un-opsonized siCEM cells was used as a negative control to set the gating for functional marker expression (**A**, top panels). (**B**–**I**) the y-axes show the frequency of adaptive NK cells (**B**–**E**) and conventional NK cells (F-I) from CMV^+^PLWH (red bars) versus CMV mono-infected (CMV^+^HIV^−^) individuals (green bars) expressing CD107a (**B**,**F**), and secreting CCL4 (**C**,**G**), IFN-γ (**D**,**H**) and TNF-⍺ (**E**,**I**). Bar colors refer to the same study groups as defined in Figure 1 panels B and C. Each data point represents results for a single individual. Bar graph heights and error bars show the median and interquartile ranges (IQR) for each data set. The significance of between-group differences was assessed using Mann—Whitney tests. *p*-values indicating the significance of between group differences are shown over the lines linking the groups being compared. Data points corresponding to results attributed to females are distinguished from those attributed to males by being shown in light green.

**Figure 5 viruses-15-00323-f005:**
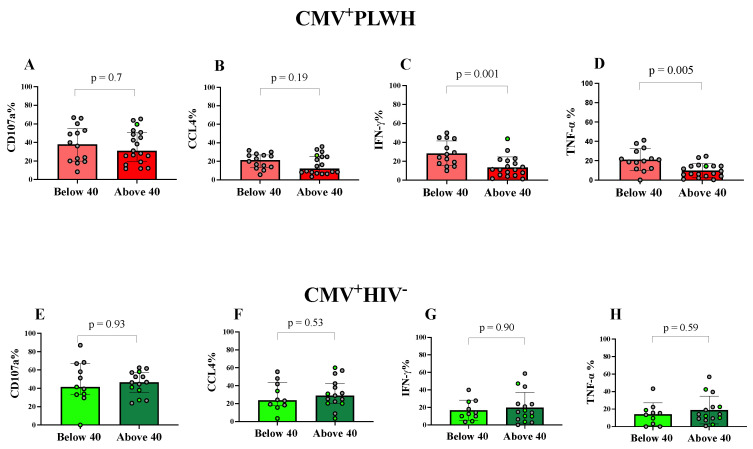
A higher frequency of adaptive NK cells from younger compared to older CMV^+^PLWH responded to anti-HIV Envelope-specific antibody opsonized HIV-infected cells by secreting IFN-γ and TNF-α. The y-axes show the frequency of adaptive NK cells (**A**–**H**) from CMV^+^PLWH (**A**–**D**) and CMV mono-infected (CMV^+^HIV^−^) individuals (**E**–**H**) aged <40 yrs or >40 yrs producing CD107a (**A**,**E**), CCL4 (**B**,**F**), IFN-γ (**C**,**G**) and TNF-α (**D**,**H**). Bar colors define the same groups as described in Figure 1 panels B and C. Each data point represents results for a single individual. Bar graph heights and error bars show the median and interquartile ranges (IQR) for each data set. The significance of between-group differences was assessed using Mann—Whitney tests. *p*-values for the comparison of adaptive NK cell frequencies between those who were <40 versus >40 yrs of age are shown over the lines linking the two populations being compared. Data points corresponding to results attributed to females are distinguished from those attributed to males by being shown in light green.

**Table 1 viruses-15-00323-t001:** Study Population Characteristics.

	CHACS		PI Cohort		HIV Uninfected	Significance of between Group Comparisons		
	CMV ^+^ PLWH (*n* = 127) Group 1	CMV ^+^ HIV^−^ (*n* = 37) Group 2	CMV ^+^ PLWH (*n* = 28) Group 3	CMV ^+^ PLWH (*n* = 15) Group 4	CMV ^+^ HIV^−^ (*n* = 22) Group 5	1 vs. 2 ^2^ 1 vs. 3 ^3^ 1 vs. 4 ^4^ 1 vs. 5 ^5^	2 vs. 3 ^6^ 2 vs. 4 ^7^ 2 vs. 5 ^8^	3 vs. 4 ^9^ 3 vs. 5 ^10^ 4 vs. 5 ^11^
Age ^1^	55.7 (51.1, 59.3)	57.7 (50.8, 62.4)	33.2 (30.8, 38.2)	54.1 (52.9, 60.1)	31.5 (25.2, 36.6)	*p* > 0.99 ^2^ *p* < 0.0001 ^3^ *p* > 0.99 ^4^ *p* < 0.0001 ^5^	*p* < 0.001 ^6^ *p* > 0.98 ^7^ *p* > 0.0001 ^8^	*p* < 0.0001 ^9^ *p* > 0.99 ^10^ *p* < 0.0001 ^11^
Sex, *n* (%) ● Male ● Female	117 (92.1) 10 (7.9)	28 (75.77) 9 (24.3) ^4^	26 (92.8) 2 (7.1)	14 (93.3) 1 (6.7)	17 (77.3) 5 (22.7)	*p* = 0.016 ^2^ *p* > 0.99 ^3^ *p* > 0.99 ^4^ *p* = 0.05 ^5^	*p* < 0.1 ^6^ *p* > 0.25 ^7^ *p* > 0.99 ^8^	*p* > 0.99 ^9^ *p* = 0.22 ^10^ *p* = 0.47 ^11^
Duration of infection (years) ^1^	Unknown	N.A.	2.2 (1.6, 2.7)	1.9 (1.8, 2.0)	N.A.			*p* = 0.15 ^9^
Viral load (HIV RNA copies/mL of plasma)	<50	N.A.	<50	<50	N.A.			
CD4^+^ count (cells/mm^3^) ^1^	575 (411, 723)	N.D.	650 (500, 926.5)	700 (600.5, 814)	N.D.	*p* = 0.14 ^3^ *p* = 0.053 ^4^		*p* > 0.99 ^9^
CD4^+^/CD8^+^ ratio ^1^	0.8 (0.54, 1.1)	N.D.	1 (0.7, 1.3)	1.1 (0.9, 1.5)	N.D.	*p* = 0.12 ^3^ *p* = 0.006 ^4^		*p* = 0.6 ^9^
Years on ART ^1^	16.0 (8.6, 19.1)	N.A.	1.4 (1.0, 2.2)	1.94 (1.8, 2.0)	N.A.	*p* < 0.0001 ^3^ *p* < 0.0001 ^4^		*p* > 0.99 ^9^

CHACS = Canadian HIV and Aging Cohort Study; PI = Primary Infection; N.A. = Not Applicable; N.D. = Not Determined; CMV^+^PLWH = Cytomegalovirus positive People Living with HIV; CMV^+^HIV^−^ = Cytomegalovirus mono-infected; vs. = versus.^1^ Median (interquartile range). The significance of differences in Age, CD4 count, CD4:CD8 ratio and Years on ART was assessed using Kruskal—Wallis tests with Dunn’s post tests. The significance of the difference in duration of infection between the two groups in the PI cohort was assessed using a Mann—Whitney test. The significance of proportional between group differences in differences sex distribution was assessed using Fishers exact tests.

## Data Availability

The datasets generated during and/or analyzed during the current study are not publicly available, but are available from the corresponding author on reasonable request.

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
