# Peer review of "The Frequency and Function of NKG2C+CD57+ Adaptive NK Cells in Cytomagalovirus Co-Infected People Living with HIV Decline with Duration of Antiretroviral Therapy"

_viruses, 2023, doi:10.3390/v15020323_

Round 1

Reviewer 1 Report (Previous Reviewer 1)

In this revised manuscript, the authors have provided substantial ameliorations, which have addressed most of my former comments.

I respectfully suggest the following minor point:

-In my former comment#5, the information in brackets (line 339 in old version) should be “Fig 5A, B, E, F” rather than “Fig 5A, B”. However in new version of manuscript, this error was not corrected (line 379 in new version).

Author Response

All changes requested have been made.

Reviewer 2 Report (New Reviewer)

Author Response

No additional changes were requested by this reviewer.

This manuscript is a resubmission of an earlier submission. The following is a list of the peer review reports and author responses from that submission.

Round 1

Reviewer 1 Report

In this manuscript, the authors collected PBMC from two HIV study cohorts and evaluated age-associated adapNK frequency and cellular functions. Results have shown that the frequency and functions of adapNK in CMV+PLWH population were negatively correlated with age. Overall, this study has clearly demonstrated the relationship between age and adapNK in CMV/HIV coinfection population through summarization of extensive data and analyses, which provided meaningful insights into the influence of viral infection and age to NK cell activity. However, the following points need to be clarified before proceeding to the consideration of publication.

1 Does the duration of ART have any impacts on the results? Based on Table 1, the participants from the CHACS cohort with the age above 40 have obtained a long term of ART (a median of 15.1 years) while the younger patients from the PI cohort have only obtained a short term of ART (a median of 1.5 years). The duration of ART application should be taken into consideration especially it has distinct different between old and younger patients.

2 Please add the fluorescent value of X- and Y-axes for flow plots (Fig 1A and Fig 3A).

3 In the second plot from left of Fig 1A, the gating for single cells is incorrect (SSC-A vs FSC-A?).

4 In Fig 3A, the lower 8 plots showing the NK factors are misleading, since their values are incompatible with the data from Fig 3B-I. For example, the representative percentage of CD107a from adapNK in Fig 3A is around 3%, while the result of the same factor in Fig 3B is around 40%.

5 In line 339, the information in brackets should be “Fig 5A, B, E, F”.

6 Some typos are occasionally seen. Please revise the whole manuscript carefully.

Reviewer 2 Report

This manuscript seeks to quantify the frequency and functional characteristics of adaptive-like NK cells in PLWH and regards to age. The cohorts selected are great and well thought out, but the interpretation and presentation of the results are severely flawed and make the conclusions hard to understand. The cohort selected in not robust enough to make major insights into age effects.  The major limitation is the selection of 40 years old as a cuttoff for age. There is no rationale except that the authors results are significant at that cuttoff. It appears the only significant finding is that PLWH have higher levels of these adaptive NK cells, which has been described. Most of the findings have already been published including Heath et al. 2016.  The authors fail to convey what the significance of their findings are for human health or HIV. 

The authors should change adapNK cell nomenclature as that is not standard. Perhaps ALNK or something else.  And this definitions should first be defined by the surface marker definition the authors are using.

The title needs to be revised.

The abstract needs to be rewritten, it is very confusing and has methods, conclusions and background all mixed in.

The authors define adaptive NK cells as NKG2C+CD57+, but what about the other NK markers. Are they CD56 bright/dim or CD3- or CD16+. This needs to be clarified.

Why was 40 years old selected. This is not typically an “older” age cuttoff for the immune system. Usually >65-70 years old is selected.

In table one the comparisons are a median of ~32 years old and ~55 years old, that is not a big age difference in terms of immunity.

How does ART impact these adaptive NK cells. What ART drugs were used for the cohort, all the same or different.

The ADNKA assay methods are not clear and need to be revised to include more detail.

In table 1, there should be a test for significance to determine if ther are significant differences in the attributes of the groups. In addition, race/ethnicity information should be provided as this can also impact immune cell phenotypes.

In table 1 all the HIV uninfected are below 40 years so how is figure 1 B determined?

The frequencies of the adaptive NK cells in some cases reach 80%, is this 80% of NK cells?  Is this in line with prior literature on the frequency of these cells.

While the 40yr. old cuttoff is not correct, the authors should not a very weak R correlation for age in figure 1D-F. This analysis should be removed from the conclusions as this is not robust.

Figure 2A should be removed, but it appears that PLWH have higher adaptive NK cells.

Figure 4, it is already known that adaptive NK cells have better function compared to conventional. What is this adding?